# Is the Adrenal Incidentaloma Functionally Active? An Approach-To-The-Patient-Based Review

**DOI:** 10.3390/jcm11144064

**Published:** 2022-07-14

**Authors:** Stella Bernardi, Veronica Calabrò, Marco Cavallaro, Sara Lovriha, Rita Eramo, Bruno Fabris, Nicolò de Manzini, Chiara Dobrinja

**Affiliations:** 1Department of Medical, Surgical and Health Sciences, University of Trieste, Strada di Fiume 447, 34149 Trieste, Italy; saralvrh@gmail.com (S.L.); b.fabris@fmc.units.it (B.F.); ndemanzini@units.it (N.d.M.); or chiara.dobrinja@units.it (C.D.); 2SS Endocrinologia, UCO Medicina Clinica, ASUGI (Azienda Sanitaria Universitaria Giuliano Isontina), Cattinara Teaching Hospital, Strada di Fiume 447, 34149 Trieste, Italy; veronica.calabro@asugi.sanita.fvg.it; 3UCO Radiologia, ASUGI (Azienda Sanitaria Universitaria Giuliano Isontina), Cattinara Teaching Hospital, Strada di Fiume 447, 34149 Trieste, Italy; mrc.cavallaro@virgilio.it; 4UCO Clinica Chirurgica, ASUGI (Azienda Sanitaria Universitaria Giuliano Isontina), Cattinara Teaching Hospital, Strada di Fiume 447, 34149 Trieste, Italy; rita.eramo@asugi.sanita.fvg.it

**Keywords:** adrenal incidentaloma, surgery, perioperative management, primary aldosteronism, autonomous cortisol secretion, pheochromocytoma

## Abstract

Adrenal incidentalomas are a common occurrence. Most of them are adrenocortical adenomas that do not cause harm and do not require surgery, but a non-negligible proportion of incidentalomas is represented by functionally active masses, including cortisol-secreting adenomas (12%), pheochromocytomas (3–6%), aldosterone-secreting adenomas (2–3%), as well as malignant nodules, such as adrenocortical carcinomas (2–5%), which can be either functioning or non-functioning. All patients with an adrenal incidentaloma should undergo a few biochemical screening and confirmatory tests to exclude the presence of a functionally active mass. In this approach-to-the-patient-based review, we will summarize current recommendations on biochemical evaluation and management of functionally active adrenal incidentalomas. For this purpose, we will present four case vignettes, whereby we will describe how patients were managed, then we will review and discuss additional considerations tied to the diagnostic approach, and conclude with practical aspects of patient perioperative management. To improve the perioperative management of patients with functional adrenal incidentalomas, multidisciplinary meetings are advocated.

## 1. Introduction

Adrenal incidentalomas are a common occurrence. The prevalence of adrenal incidentaloma ranges from 4% to 10% in radiological studies and from 1% to 8.7% in autopsy series [1,2]. Adrenal incidentalomas increase in frequency with age: they are uncommon in patients younger than 30 years (0.2%) as compared with patients aged more than 70 years (7%) [3]. In case of an adrenal incidentaloma, two issues arise: whether the lesion is potentially malignant and whether it is functionally active, as both scenarios require surgery.

It has been shown that most incidentalomas are non-functioning adrenocortical adenomas that do not cause harm and do not require surgery (occasionally, they can be also myelolipomas, hamartomas, or granulomatous infiltrations of the adrenal gland). According to the American College of Radiology (ACR) [2], diagnostic imaging features of the benignity of adrenal masses are those consistent with the presence of fat, such as CT attenuation ≤10 Hounsfield units (HU); signal intensity loss on the MRI between in-phase and opposed-phased T1-weighted gradient-echo images; and lack of enhancement on the CT (change of <10 HU between pre- and post-contrast imaging). Based on the ACR and the ESE/ENSAT guidelines [2,4], masses without benign imaging features are candidates for further imaging and/or surgery. However, size and growth are also important variables in predicting malignancy, and they are generally used to make decisions regarding surgery or further imaging [2]. For instance, surgery may be considered in case an adrenal mass with benign imaging features measures >4 cm [4], and it is recommended in case an adrenal mass without benign imaging features measures >4 cm [2,4]. Size cutoff is considered unreliable to be used alone as a criterion for malignancy; nevertheless, adenomas comprise the vast majority of masses measuring less than 4 cm, while they represent only 18% of masses above 6 cm [5]. By contrast, adrenocortical carcinomas represent 2% of all masses measuring less than 4 cm in diameter, 6% of masses measuring between 4 and 6 cm, and 25% of masses greater than 6 cm [5,6,7].

Besides imaging features, the second aspect that should be taken into account is the endocrine activity of the adrenal incidentaloma. This is key to decide whether the mass should be surgically removed, as well as to tailor perioperative patient management. ESE/ENSAT guidelines recommend that all patients with an adrenal incidentaloma should be clinically assessed for signs consistent with the presence of a functionally active adrenal mass [4], including hypertension, hypokalemia, diabetes mellitus, asymptomatic vertebral fractures, as well as hirsutism, virilization, or gynecomastia. In addition, it is important to pay attention to other signs, such as easy bruising, proximal myopathy, facial plethora, and striae, which are the features that best discriminate Cushing’s syndrome [8,9]. Then, all patients should undergo the exclusion of autonomous cortisol secretion by the 1 mg overnight dexamethasone suppression test, and the exclusion of pheochromocytoma by measurement of plasma-free metanephrines or urinary fractionated metanephrines. Only in patients with hypertension and/or hypokalemia [4,10] should the presence of an aldosterone-secreting tumor be excluded with the aldosterone-to-renin ratio (ARR), while sex hormones should be measured in patients with signs of androgen or estradiol excess [4].

Notwithstanding these recommendations, a recent systematic review of the literature shows that only a minority of patients undergo biochemical testing (a median of 18% IQR 15–28%) [11]. This approach-to-the-patient-based-review summarizes current recommendations on the biochemical evaluation and management of functionally active adrenal incidentalomas. For this purpose, we will present four case vignettes [12], whereby we will describe the approach to the patient that we chose, then we will review and discuss additional considerations tied to the diagnostic approach, and the perioperative management of each scenario.

## 2. Primary Aldosteronism

### 2.1. Case 1

A 48-year-old man came to the endocrine service because a recent CT scan of the abdomen showed an incidentaloma in the right adrenal gland measuring 1 cm and with low attenuation (<10 HU) (Figure 1a). The patient had a background of hypertension and diabetes, and two traumatic vertebral fractures. Most importantly, he had undergone a left subtotal nephrectomy for a renal carcinoma almost three years before, for which he attended periodic CT scans. Given his history of extra-adrenal malignancy, we reviewed his previous CT scans with a radiologist, and found that the adrenal nodule was already present two years before and was unchanged. Given that the patient had a 1 cm adrenal nodule with benign imaging features and documented stability, further imaging was not indicated and we prescribed laboratory tests (Table 1 summarizes recommended screening tests).

The patient was switched to a calcium channel blocker and doxazosin for the 4 weeks prior to the laboratory tests, which showed that aldosterone was 26.4 ng/dL (reference ranges 1.5–15 ng/dL), renin was suppressed, being 0.5 µU/mL (reference ranges 2.8–40 µU/mL), and potassium was 3.53 mEq/L. The aldosterone-to-renin-ratio (ARR) was 559 ng/dL/mIU/L as assessed with the ARR-App [13], whereby values <2.06 ng/dL/mIU/L are the cutoff to exclude primary aldosteronism. It has to be noted that the ARR is not always easily estimated, due to different ways of measuring it (assays and units). To overcome this issue, the ARR-app is a recently introduced tool, which is endorsed by the Italian Society of Arterial Hypertension [10], which provides a calculation of the ARR regardless of the units of measure and also taking into account potassium levels [13]. The ARR of our patient was consistent with the presence of a primary aldosteronism. Otherwise, post-dexamethasone cortisol was appropriately suppressed, being 0.74 µg/dL (values <1.8 µg/dL exclude autonomous cortisol secretion), and 24 h urinary fractionated metanephrines were below normal ranges.

Then, before surgery, we scheduled an adrenal venous sampling (AVS). For this sampling, the adrenal and peripheral (either inferior vein cava or femoral) veins are sampled to measure cortisol and aldosterone [15]. The procedure can be done with or without cosyntropin stimulation. First of all, it is expected that the adrenal vein has more cortisol than a peripheral vein; therefore a properly cannulated adrenal vein should show an adrenal vein-to-peripheral vein cortisol ratio ≥2 under unstimulated conditions, or ≥3 during cosyntropin stimulation [15]. This ratio is also called “selectivity index” (SI). In our patient, we scheduled the AVS without cosyntropin stimulation and the SI was 196 on the right adrenal vein, and 101 on the left adrenal vein. Secondly, aldosterone levels (A) are normalized to corresponding cortisol levels (C) in each side (A/C). Then, each side A/C is divided by the other to produce the lateralization index (LI). A LI ≥2 under unstimulated conditions and ≥2.6 under stimulated conditions is consistent with a unilateral source [15]. In our case the patient exhibited a right lateralization (LI of 53 right/left).

The saline infusion test confirmed the presence of primary aldosteronism, because aldosterone was unresponsive to water load, as after infusion of 2 L of saline in 4 h, aldosterone was 19.8 ng/dL (>10 ng/dL is a sign of very probable primary aldosteronism) [14].

Once AVS indicates secretion laterality, patients can be surgically treated. In the case of our patient, in the weeks preceding surgery, spironolactone was added to the antihypertensive therapy. Then, the patient underwent a laparoscopic right adrenalectomy, and the pathology confirmed the removal of an adrenal adenoma measuring 10 × 9 × 12 mm, consistent with the diagnosis of Conn’s disease. After surgery, spironolactone was discontinued and the patient did not need steroid replacement therapy. Potassium was 4.95 mEq/L. Over time, antihypertensive therapy was further reduced and blood pressure improved, although it did not completely normalize. Aldosterone was 4.1 ng/dL and renin was 2.5 µU/mL 6 months after surgery.

### 2.2. Diagnosis and Subtype Classification of Primary Aldosteronism

The detection of primary aldosteronism relies on ARR values. ARR can be calculated also with an application, called the ARR-app, which is a tool that enhances the detection rate of primary aldosteronism [13]. High baseline ARR values imply very high positive likelihood ratio of this condition [16]. It has to be noted that measurement of aldosterone and renin requires adequate patient preparation [14]. Plasma potassium should be within reference ranges, as hypokalemia reduces aldosterone levels and ARR, while sodium intake should not be restricted as hyponatremia increases aldosterone levels and ARR. The measurement of potassium and sodium in a 24 h urine collection is therefore recommended for a proper interpretation of ARR [10]. Agents that markedly affect the ARR should be withdrawn for at least 4 weeks before, these include spironolactone, potassium-wasting diuretics, products containing liquorice, as well as beta-blockers, renin-angiotensin inhibitors, diuretics, and NSAID. Calcium channel blockers and doxazosine can be used to control blood pressure before ARR measurement [10].

Glucocorticoid cosecretion is frequently found in primary aldosteronism and contributes to associated metabolic risk [17]. The overproduction of mineralocorticoids and glucocorticoids is referred as Connshing syndrome and it is independent of primary aldosteronism subtype or tumor tissue genotypes. Generally, glucocorticoid excess in patients with primary aldosteronism is at least similar to that of subclinical Cushing’s syndrome [17]. In patients with high ARR, it is important to rule out autonomous cortisol secretion by the 1 mg overnight dexamethasone suppression test. The presence of autonomous cortisol can impact perioperative patient management and postoperative hypocortisolism, as well as medical therapy in patients who cannot be operated on.

It is recommended that patients with a positive ARR undergo one or more confirmatory tests to definitely confirm or exclude the diagnosis of primary aldosteronism [14]. The saline infusion and the captopril challenge are the most widely used confirmatory tests [18]. In the case of a saline infusion test, an infusion of 2 L of saline over 4 h should lower aldosterone levels <5 ng/dL, whereas levels >10 ng/dL are a sign of very probable primary aldosteronism. In case of the captopril challenge test, plasma aldosterone is normally suppressed >30%. Nevertheless, in the setting of spontaneous hypokalemia or very high ARR values, there is no need of these tests [10,14,16].

Once the diagnosis of primary aldosteronism has been made, the evidence of an adrenal incidentaloma is not enough to surgically remove the nodule, except for patients aged <35 years, with spontaneous hypokalemia, marked aldosterone excess, and a unilateral adrenal lesion consistent with a cortical adenoma [14]. The rationale behind this is that the presence of an adrenal incidentaloma, which is increasingly more common with age [3], does not exclude that primary aldosteronism is sustained by one or more smaller and CT-invisible nodules on the opposite adrenal gland. It has been shown that patients with primary aldosteronism may have aldosterone-producing cell clusters and/or secondary nodules expressing aldosterone synthase in association with an aldosterone-producing adenoma [19]. Therefore, patients who might be surgically treated should undergo an adrenal venous sampling to distinguish secretion laterality before surgery. A recent multicenter study has shown that patients diagnosed by CT have a decreased likelihood of achieving complete biochemical success after surgery as compared with a diagnosis made by adrenal vein sampling [20]. It is conceivable that in the future, specific positron emission tomography radiotracers might have a role in the subtype evaluation of aldosterone-producing adenomas [21].

### 2.3. Perioperative Management of Primary Aldosteronism

Overall, for unilateral primary aldosteronism, the standard of care in terms of safety and feasibility is AVS-guided unilateral trans-peritoneal or retro-peritoneal laparoscopic adrenalectomy, which is associated with complete biochemical success in more than 98% of cases and either cures or has partial clinical success in terms of lowering blood pressure in some 80% of cases [10,20]. Partial adrenalectomy, which preserves remnant adrenal function and avoids adrenal insufficiency, is not routinely recommended given the risk of having different aldosterone-producing cell clusters [19], and it should be only performed under the guidance of super-selective adrenal vein sampling [10]. Medical therapy is recommended in order to normalize blood pressure and potassium in patients with bilateral disease, or in those who are not candidates for adenalectomy [10].

Before surgery, blood pressure and potassium must be corrected with the use of mineralocorticoid receptors antagonists (MRA) and/or potassium supplementation. This is important to avoid hypertension and hypokalemia during surgery. It has been shown that intraoperative systolic and diastolic blood pressure values increase as compared to the preoperative ones and that intraoperative potassium levels decrease by 0.7 mmol/L after 30 min of pneumoperitoneum induction [22].

On postoperative day 1, MRA and potassium should be withdrawn, and antihypertensive therapy should be reduced, if appropriate. During the first weeks after surgery clinicians should recommend a generous sodium diet, which helps avoid hyperkalemia. Potassium should be measured after surgery. However, persistent postoperative hypoaldosteronism with hyperkalemia, requiring long-term fludrocortisone treatment, is rare, as it occurs in 5% of adrenalectomized patients with primary aldosteronism, [23]. Impaired kidney function is the main predictor of this occurrence [23]. Aldosterone and renin should be measured 1 and 6 months after surgery to confirm biochemical cure [10,20].

## 3. Cortisol-Secreting Adenoma

### 3.1. Case 2

A 59-year-old woman came to the endocrine service because a recent CT scan of the abdomen showed an incidentaloma in the left adrenal gland measuring 2 cm and with low attenuation (<10 HU), as shown in Figure 1b. The patient suffered from nephrolithiasis, which was the reason why she underwent a CT scan of the abdomen, and had a background of resistant hypertension only. Her last ambulatory blood pressure monitoring (ABPM) showed overall blood pressure values of 163/93 mmHg with mean day BP values of 172/100 mmHg and mean night BP values of 137/78 mmHg, while taking four antihypertensive drugs. Given that the patient had a 2 cm adrenal nodule with benign imaging features, further imaging was not indicated, but we prescribed further laboratory tests (Table 1).

Post-dexamethasone cortisol was 14.9 µg/dL (values > 5 µg/dL indicate the presence of an autonomous cortisol secretion), 24 h urinary free cortisol was increased, as it was 566.5 µg/24 h (reference range is 20.9–292.3 µg/24 h), and ACTH was 4.5 ng/L (reference range is 7.2–63.3 ng/L). Otherwise, 24 h urinary metanephrines were below normal ranges and the ARR < 2 excluded primary aldosteronism. Taking into account the clinical and biochemical parameters, the patient was diagnosed with an ACTH-independent autonomous cortisol secretion [4].

Given the presence of autonomous cortisol secretion and resistant hypertension, the patient underwent laparoscopic left adrenalectomy and final pathology was consistent with an adrenal cortical adenoma measuring 32 × 24 × 20 mm. Morphological analysis (Weiss system [24]) showed small cells with round nuclei and no nucleoli, almost no mitoses, >25% clear cells, no venous, sinusoidal invasion or capsular infiltration. Ki67 staining showed a proliferation index <5%. Our patient received hydrocortisone (100 mg iv) during surgery and cortone acetate 25 mg/day after surgery that was progressively tapered. On postoperative day 1, basal cortisol was 2.6 µg/dL. In addition, we prescribed low molecular weight heparin (LMWH) for 4 weeks. One year after surgery, the low dose ACTH test showed adrenal insufficiency resolution, as baseline cortisol was >5 µg/dL and stimulated cortisol levels were >22 µg/dL. At that point, steroid therapy was withdrawn. Over time, blood pressure significantly improved, although it did not completely normalize.

### 3.2. Diagnosis of Cortisol-Secreting Adenoma

In patients with an adrenal incidentaloma, it is recommended to evaluate the presence of autonomous cortisol secretion by the 1 mg overnight dexamethasone suppression test. In particular, post-dexamethasone cortisol between 1.9 and 5 μg/dL should be considered as evidence of “possible autonomous cortisol secretion” and >5 μg/dL should be taken as evidence of “autonomous cortisol secretion” [4]. Then, it is recommended to confirm the autonomous cortisol secretion by measurement of 24 h urinary free cortisol and/or late-night salivary cortisol, and to prove ACTH-independency by ACTH measurement [4].

In a recent work by Araujo-Castro et al., post-dexamethasone cortisol was the best predictor of autonomous cortisol secretion development over time, and the best threshold to predict it was 1.4 μg/dL (sensitivity 59% and specificity 72%) [25]. Therefore, in cases where post-dexamethasone cortisol is >1.4 μg/dL, it might be taken into account to repeat the 1 mg overnight dexamethasone suppression test annually for up to 5 years [25].

### 3.3. Perioperative Management of Cortisol-Secreting Adenoma

In patients with an ACTH-independent autonomous cortisol secretion, ESE/ENSAT guidelines suggest an individualized therapeutic approach. Age, comorbidities, and patient’s preferences should be taken into account before considering surgical intervention, given that surgery may not normalize or improve patient clinical phenotype. There is an indication of surgery in a patient with post-dexamethasone cortisol >5 μg/dL and the presence of at least two comorbidities potentially related to cortisol excess, of which at least one is poorly controlled by medication [4]. It goes without saying that surgery is also the first-line therapeutic option in case of an overt Cushing syndrome of adrenal origin [26].

Endocrine Society guidelines recommend initial resection of primary lesion(s), unless surgery is not possible, by an experienced surgeon [26]. Before surgery, it is important to treat cortisol-dependent comorbidities (such as diabetes, hypertension, hypokalemia, infections). In addition, patients should be evaluated for risk factors of venous thrombosis, and in those undergoing surgery, it is recommended perioperative prophylaxis for venous thromboembolism [27]. Anticoagulation treatment should be considered especially in the 4 weeks after surgery [26]. In patients with autonomous cortisol secretion, it is possible (advisable in case of Cushing syndrome) to normalize cortisol levels with steroidogenesis inhibitors, such as metyrapone, as demonstrated in a recent study showing that metyrapone (750–1000 mg daily) was effective in normalizing biochemical and clinical parameters before surgical intervention, with minimal side effects [28]. Metyrapone should be withdrawn 3 days before surgery.

It is well known that in the case of ACTH-independent autonomous cortisol secretion, the surgical removal of the adrenal mass causing hypercortisolism is followed by postoperative insufficiency of the remaining adrenal, which requires glucocorticoid replacement therapy for a variable length of time [29,30]. Interestingly, it seems that post-dexamethasone cortisol < 1.2 µg/dL (before surgery) could rule out the possible occurrence of hypocortisolism in patients undergoing the removal of an adrenal adenoma [31]. Nevertheless, in all the other cases, precautionary steroid therapy to protect from adrenal failure includes: 50–100 mg of hydrocortisone (diluted in 250 mL of saline) given intravenously during surgery, 50 mg of hydrocortisone i.v. + 25 mg of hydrocortisone i.v. on postoperative day 1, and 25 mg of cortone acetate + 12.5 mg of cortone acetate on postoperative day 2 [30,32]. This replacement therapy should be progressively reduced and then discontinued when the response to the ACTH stimulation test shows recovery of the hypothalamic-pituitary-adrenal axis [26]. It has been reported that in 98% of patients, adrenal function recovers after 6.5 months from surgery; nevertheless, a longer period (up to 4 years) may be required in a few cases [29].

## 4. Pheochromocytoma

### 4.1. Case 3

A 62-year-old man came to the endocrine service because a recent CT scan of the abdomen showed an incidentaloma in the left adrenal gland measuring 5 cm and with high attenuation (>10 HU), as shown in Figure 1c. The CT scan was performed at the A&E service to investigate the recent onset of abdominal pain, which was found to be related to an acute cholelithiasis, for which he was admitted to the surgical department. The patient reported the recent onset of hypertension for which he was taking an angiotensin-receptor blocker. The radiological characteristics of the adrenal incidentaloma (size, attenuation on noncontrast CT) were consistent with a potentially malignant mass, but before prescribing additional radiological exams, we recommended laboratory tests to ascertain if the lesion was functionally active (Table 1).

In this case, 24 h urinary fractionated metanephrines measured by liquid chromatography mass spectrometry were repeatedly found elevated. Metanephrine was 723.8 µg/24 h in the first 24 h urinary collection, and 931.2 µg/24 h in the second one (reference range 44–261 µg/24 h), whereas normetanephrine was 823.2 µg/24 h and 374 µg/24 h (reference range 138–521 µg/24 h). Otherwise, post-dexamethasone cortisol was 1.1 µg/dL and ARR was <2, excluding autonomous cortisol secretion as well as primary aldosteronism. These results were consistent with the presence of a pheochromocytoma on the left adrenal gland. The patient antihypertensive therapy was changed to doxazosine.

Adrenal surgery is the only available treatment for pheochromocytomas. Before scheduling adrenalectomy, we measured calcitonin, calcium, and PTH, to exclude clinically relevant MEN2 associated disorders, and they were all within reference ranges. The patient underwent genetic testing, which excluded *RET*, *TMEM127*, *MAX*, *VHL*, *SDHA*/*B*/*C*/*D* mutations, indicating a sporadic pheochromocytoma. In addition, due to the size of the adrenal mass, in order to exclude multisite or metastatic disease, we scheduled a ^123^I-metaiodobenzylguanidie (MIBG) scintigraphy, which showed only adrenal MIBG uptake (Figure 1d).

The patient underwent laparoscopic left adrenalectomy, and final pathology confirmed the presence of a pheochromocytoma, measuring 55 × 45 × 25 mm. After surgery, blood pressure improved and 24 h urinary metanephrines were normal at the annual follow-up.

### 4.2. Diagnosis of Pheochromocytoma

In patients with an adrenal incidentaloma, it is recommended to ascertain the presence of a pheochromocytoma by measuring plasma-free or urinary fractionated metanephrines [4,33]. Both measurements have high diagnostic sensitivity but suboptimal diagnostic specificity, meaning that false-positive results can be common [34]. Nevertheless, an increase in both metanephrine and normetanephrine, or an increase in fractionated metanephrines 3-fold or more above upper cutoff limits, are rarely due to a false-positive result [33,35]. In cases of borderline positive tests, it is useful to adopt a wait-and-retest approach. It has to be noted that a few medications may cause falsely elevated results, and they should be discontinued before measuring fractionated metanephrines. These drugs include: acetaminophen, labetalol, sotalol, methyldopa, tryciclic antidepressants, buspirone, phenoxybenzamine, monoamine oxidase inhibitors, sympatomimetics, sulphasalazine, and levodopa, as well as cocaine.

Before adrenalectomy, preoperative exams should screen also for the presence of genetic disease, and/or metastatic or multifocal disease. Genetic disease seems to account for up to 40% of the cases [36,37,38] and it can predispose to bilateral pheocromocytoma. Genetic disease includes MEN2A, MEN2B, von-Hippel Lindau (VHL), neurofibromatosis type 1 (NF1), as well as mutations in the genes of myc-associated factor-X (*MAX*), transmembrane protein 127 (*TMEM127*) [38], and the succinate dehydrogenase complex (*SDHx*). Dopaminergic, noradrenergic, and adrenergic phenotypes should be used to establish the priorities for specific testing [33]. Genetic testing can help choose the correct surgical approach. In particular, patients with pheochromocytomas related to some of the above-mentioned genetic syndromes should be offered a cortical-sparing (partial) adrenalectomy to maximize adrenal function [39,40]. These include patients with a *TMEM127* mutation or VHL disease that infrequently progress to metastases [38]. By contrast, an open approach (and consideration for lymphadenectomy) may be preferred in patients with *SDHB* mutations, who have a higher rate of metastatic or multisite disease [39].

In patients with pheochromocytomas and an increased risk for metastatic or multifocal disease, due to the large size of the primary tumor or to an extra-adrenal location, it is suggested the use of ^123^I-metaiodobenzylguanidie (MIBG) scintigraphy. ^18^F-FDG PET/CT scanning is the preferred imaging modality in patients with known metastatic pheochromocytomas [33].

### 4.3. Perioperative Management of Pheochromocytoma

Guidelines recommend that all patients with a functionally active pheochromocytoma undergo preoperative α-adrenergic receptor blockade for at least 7–14 days prior to surgery. Calcium channel blockers can be used as add-on drugs to α-adrenoceptor blockade in order to control blood pressure levels [41]. In patients presenting with tachycardia or tachyarrythmias requiring the use of β-blockers, β-adrenoceptor blockade can be introduced only after introduction of proper α-adrenoceptor blockade [41]. In addition, during the preparation period, it is recommended to prescribe a high-sodium diet and high fluid intake to prevent hypovolemia due to adrenergic tone with pressure natriuresis [33]. During surgery, to control the adrenergic tone, it is recommended to infuse saline, and to use urapidil or labetalol. Important principles also include minimal manipulation of the tumor to prevent catecholamine release with resultant hemodynamic instability and/or tumor rupture, as well as early ligation of the adrenal vein [39]. During the postoperative period, it is important to control blood pressure, as well as glucose (and electrolyte) levels. In a recent work on 159 patients with pheochromocytoma, 19% of them had postsurgical complications. Prolonged hypotension was the most common, followed by hypoglycemia, and they were more common in patients with diabetes, cerebrovascular disease, higher urinary metanephrines, or masses >5 cm [42]. Although the relative reduction in catecholamines increases the risk of hypoglycemia, in the cases of hypotension and hypoglycemia, the occurrence of hypoadrenalism also has to be taken into account. Fractionated metanephrines should be checked after 1 month and then annually to exclude recurrent or metastatic disease [33].

## 5. Potentially Malignant Adrenal Mass

### 5.1. Case 4

A 46-year-old man came to the endocrine service because a recent CT scan of the chest and abdomen showed an incidentaloma in the right adrenal gland measuring 9.3 cm with features consistent with a potentially malignant mass (Figure 1e). He had no history of malignancy, but reported a recent spontaneous rib fracture and weight loss. DEXA showed lumbar T score of −2.6. Otherwise, he had no hypertension, diabetes, hypokalemia, or gynecomastia. We prescribed further laboratory (Table 1) and imaging exams.

In this patient, post dexamethasone cortisol was 3.2 µg/dL, indicating a possible autonomous cortisol secretion (4). Additionally, 24 h urinary free cortisol was increased, as it was 478 nmol/24 h (reference range 58–306), and ACTH was 4.8 pg/mL (4.2–48.8), consistent with cortisol secretory autonomy. Otherwise, 24 h urinary fractionated metanephrines were below normal ranges, and ARR was not measured because the patient did not have hypertension.

Before surgery, the patient underwent a ^18^F-FDG-PET CT to exclude the presence of malignant metastatic or multifocal disease, which showed no extra-adrenal ^18^F-FDG uptake (Figure 1f). The patient underwent right laparotomic adrenalectomy and final pathology showed an adrenal mass measuring 3.5 × 7.5 × 9.5 cm with a necrotic core of 6 cm containing hemorragic areas and no viable cells, surrounded by a peripheral rim of zona fasciculata. The complete absence of viable cells did not allow any morphological analysis nor any immunohistochemical staining. The exam was consistent with either an adrenal hematoma, or a nodule with necrotic-hemorragic content. Our patient received hydrocortisone (100 mg iv) during surgery and cortone acetate 25 mg/day after surgery. On postoperative day 1, basal cortisol was 0.4 µg/dL. In addition, we prescribed LMWH for 4 weeks. Six months after surgery, the patient is still under replacement therapy with low doses of cortone acetate, as baseline cortisol is 6.2 µg/dL but stimulated cortisol levels are 14 µg/dL.

### 5.2. Perioperative Management of a Potentially Malignant Adrenal Mass

Based on the ESES/ENSAT guidelines [43], an adrenal incidentaloma at increased risk of malignancy is a mass with: (i) multiple hormonal, steroid precursor, or sex hormone oversecretions; (ii) intratumoral radiological signs of malignancy (>10 HU and/or diameter greater than 6 cm); (iii) evidence of local invasion, suspected metastatic lymph nodes, distant metastasis, and/or high ^18^F-FDG-PET CT uptake. In this case, it is important to exclude the diagnosis of pheochromocytoma, and exclude the presence of autonomous cortisol secretion, as already described. In order to define the adrenal mass, imaging should include a thoracoabdominal CT with contrast injection and ^18^F-FDG-PET. An MRI with gadolinium enhancement helps when the diagnosis is doubtful, or there are suspected vascular invasion or liver metastases [43]. A preoperative biopsy of suspected ACC is not recommended [43].

In these cases, surgery should be performed by experienced surgeons (>15 adrenalectomies per year, open and laparoscopic, as well as benign and malignant) [43]. Resective surgery strongly influences the long-term prognosis [44]. Given that the only chance for a cure in case of an adrenocortical carcinoma is a complete primary tumor resection, with no tumor capsule rupture or spillage of tumor cells, it is recommended to perform an *en bloc* resection of the adremal tumor with the peritumoral/periadrenal retroperitoneal fat. Although open adrenalectomy is the standard of surgical care [43], in patients with ACC stage I-II, there are no significant differences in terms of overall recurrence rate, time to recurrence, and cancer-specific mortality between the open and laparoscopic approaches [45].

## 6. Conclusions

Incidental adrenal masses include a broad range of diseases and they always require biochemical testing, not only to evaluate if they need to be surgically removed, but also to choose the most appropriate surgical approach and perioperative management. Guidelines recommend screening adrenal function by measuring ARR, post-dexamethasone cortisol, and fractionated metanephrines. Notwithstanding these indications, most patients do not receive an initial biochemical evaluation. Multidisciplinary meetings, including those with endocrinologists, surgeons, radiologists and pathologists, improve the perioperative management of functional adrenal tumors [32,46], allowing the selection the best diagnostic and therapeutic approaches for every patient.

## Figures and Tables

**Figure 1 jcm-11-04064-f001:**
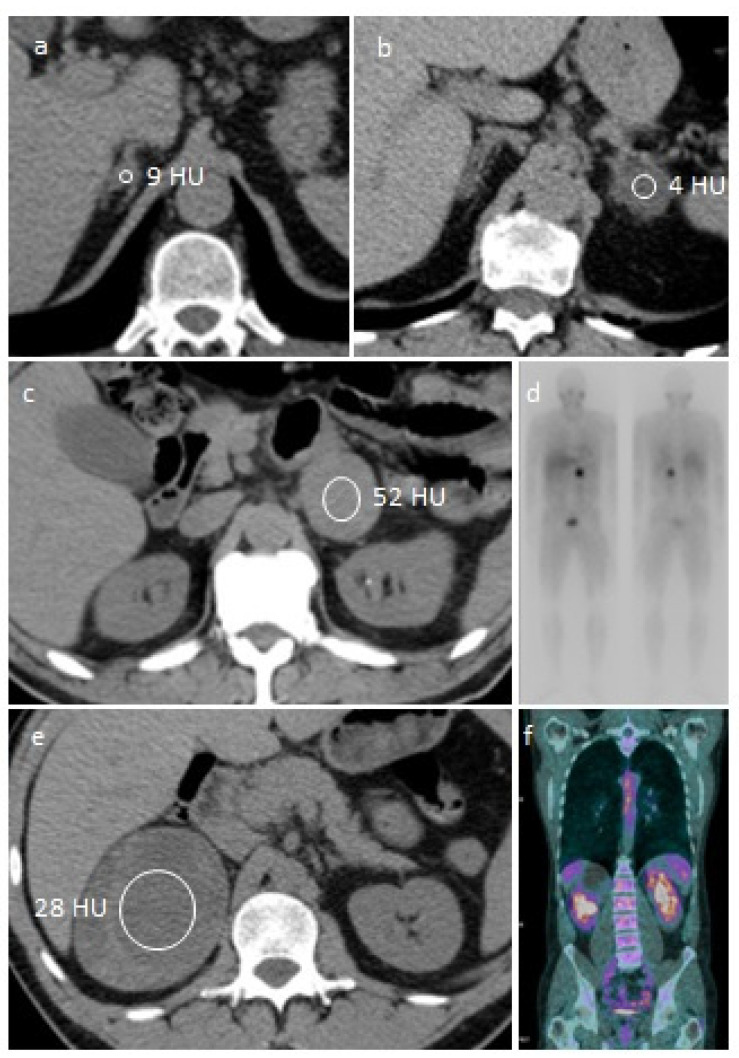
Representative images of the case vignettes. (**a**) CT scan of Case 1: unenhanced CT images show a small lesion with a low attenuation value in the right adrenal gland. (**b**) CT scan of Case 2: unenhanced CT images show a small lesion with a low attenuation value in the left adrenal gland. (**c**,**d**) CT scan and MIBG scintigraphy of Case 3: unenhanced CT image shows a high attenuation lesion in left gland (**c**) with only adrenal MIBG uptake on subsequent scintigraphy (**d**). (**e**,**f**) CT scan and ^18^F-FDG PET CT of Case 4: unenhanced CT image shows a very large inhomogeneous lesion in right gland (**e**), without extra-adrenal F-FDG uptake on subsequent PET-CT (**f**).

**Table 1 jcm-11-04064-t001:** Laboratory tests to identify functionally active adrenal incidentalomas.

Condition	Screening Tests	Confirmatory Tests	Next Steps
Primary aldosteronism	ARR	Saline infusion test Captopril challenge test	Subtype classification by adrenal vein sampling
Autonomous cortisol secretion	1 mg overnight dexamethasone suppression test	24-h urinary cortisol LNSC; ACTH	
Pheochromocytoma	Fractionated methanefrines (urine or plasma)	Not routinely performed	Genetic testing

ACTH is adrenocorticotropic hormone; ARR is aldosterone-to-renin ratio; LNSC is late-night salivary cortisol.

## Data Availability

Not applicable.

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
