# Peer review of "Is the Adrenal Incidentaloma Functionally Active? An Approach-To-The-Patient-Based Review"

_jcm, 2022, doi:10.3390/jcm11144064_

Round 1

Reviewer 1 Report

Comments to the Authors of the study entitled: “Is the adrenal incidentaloma functionally active? An approach-to-the-patient-based review”.

Authors described four cases of patients with different hormonally active adrenal tumors which served as a basis for further deliberations about appropriate diagnostic and therapeutic schemes and procedures. The literature review combined with the Authors’ own experience in the form of clinical cases was very interesting.
Nonetheless, the manuscript has numerous limitations. Some of the presented cases lack crucial clinical data, for instance there is no information regarding the potassium concentration and pre-operative treatment in a patient suffering from primary hyperaldosteronism (case 1). In case 2, there is no clear indication whether the patient was diagnosed with mild autonomous hypercortisolemia or overt Cushing’s syndrome – no data related to clinical signs and symptoms was provided; in addition, what were the concentrations of DHEA-S and LNSC in this patient?
As the Authors discussed hypercortisolemia secondary to the adrenal adenoma, they did not differentiate between the overt Cushing’s syndrome and mild autonomous hypercortisolemia – a difference critical for providing the optimal management. The Authors stated: “There is an indication of surgery in a patient with post-dexamethasone cortisol >5 ug/dL and the presence of at least two comorbidities potentially related to cortisol excess, of which at least one poorly controlled by medication” – this recommendation applies only to patients displaying mild autonomous hypercortisolemia and all patients with overt Cushing’s syndrome require urgent treatment.

Author Response

We thank the Reviewer for their valuable feedback, which has helped us improve the quality of the manuscript. Please find the point-by-point response below and the corrected copy of our manuscript, where changes have been highlighted in red link throughout the manuscript.

1. Nonetheless, the manuscript has numerous limitations. Some of the presented cases lack crucial clinical data, for instance there is no information regarding the potassium concentration and pre-operative treatment in a patient suffering from primary hyperaldosteronism (case 1).

Reply: Taking into account the reviewer comment we have added the requested clinical data, such as potassium at presentation (lines 111-112) and after surgery, as well as details on the antihypertensive therapy that we prescribed (lines 141-142).

2. In case 2, there is no clear indication whether the patient was diagnosed with mild autonomous hypercortisolemia or overt Cushing’s syndrome – no data related to clinical signs and symptoms was provided; in addition, what were the concentrations of DHEA-S and LNSC in this patient?

Reply: We have now specified that the patient suffered from nephrolithiasis and resistant hypertension only. So, taking into account clinical and biochemical parameters, based on the ESE/ENSAT guidelines, the patient was diagnosed with an ACTH-independent autonomous cortisol secretion (lines 238-240). Concentrations of LNSC were not measured as autonomous cortisol secretion was assessed with post-dexamethasone cortisol and 24 h urinary cortisol. Concentrations of DHEA-S were not measured as the patient did not exhibit signs of androgen excess to measure it.

3. As the Authors discussed hypercortisolemia secondary to the adrenal adenoma, they did not differentiate between the overt Cushing’s syndrome and mild autonomous hypercortisolemia – a difference critical for providing the optimal management. The Authors stated: “There is an indication of surgery in a patient with post-dexamethasone cortisol >5 ug/dL and the presence of at least two comorbidities potentially related to cortisol excess, of which at least one poorly controlled by medication” – this recommendation applies only, to patients displaying mild autonomous hypercortisolemia and all patients with overt Cushing’s syndrome require urgent treatment.

Reply: we have now added that “It goes without saying that surgery is also the first-line therapeutic option in case of an overt Cushing syndrome of adrenal origin [26]”, with a reference to the guidelines of the Endocrine Society on the treatment of Cushing’s Syndrome (lines 273-274).     

Reviewer 2 Report

Thank you for the opportunity to review the manuscript “Is the adrenal incidentaloma functionally
active? An approach-to-the-patient-based review by Stella Bernardi et al.

Analysis of a diagnostic and treatment path in case of adrenal incidentalomas is very important as it
becomes a more common issue in clinical endocrine practise. The manuscript is valuable although

there are some
points that need clarification. Please find my comments below.
1.
Please use pheochromocytoma instead of pheocromocytoma.
2.
Line 47: lack of enhancement (change of <10 HU 47 between pre-and postcontrast
imaging).
- it should be stated that this relates to CT imaging.
3.
Lines 52-53: By contrast, surgery may be considered in case an adrenal mass measures > 4
cm [4], and it is recommended in case an adrenal mass without
diagnostic benign imaging
features measures > 4 cm [2,4].
- shouldnt it be also <4 cm? If the adrenal mass does not
present the benign features or even the radiological features of mal
ignancy are present the
indications for surger
y include also tumours <4 cm. Maybe I do not understand correctly the
context, please clarify.

4.
Lines 65-66: including hypertension (persistent, paroxysmal or induced by medication),
hypokalemia, diabetes mellitus, asymptomatic vertebral fractures, as well as hir
sutism,
virilization or gynecomastia
- lack of other common clinical characteristics of
hypercortisolaemia.

5.
Line 234: LMWH- please describe the abb.
6.
Patient 2 and 4- what was the detailed histopathology of the tumours, according to Weiss/
Weiss
revisited criteria or other adequate system?
7.
In my opinion the information in the paragraph: lines 272-276 is not very relevant. I
understand that it was confir
med in the ref. However, this is the clear data from
understanding
the adrenal tumours biology. If this paragraph is to be left, I would
recommend
adding a comment that the cited ref. just confirm the highly suspected clinical
scenario
.
8.
Lines 303 and 381-382: the reference range to exclude autonomous cortisol secretion was
presented before, there is no need to repeat the information.

9.
Line 234: maybe cocaine, as its use mainly refers to drug abuse, may be presented at the end
of
the list of medications.
10.
Lines 351-352: In our patient, due to the size of the adrenal mass, the exam showed uptake
of the tracer on the
left adrenal gland only.- unclear.
11.
Line 410: In these cases, surgery should be performed by experienced surgeons (>15
adrenalecto
mies per year).- please add ref.

12. Line 411: Given that the only chance for cure in case of an adrenocortical carcinoma is
complete primary tumor resection, with no tumo
r capsule rupture- it is not always true. In
paediatric adrenal tumors, due to fragile capsule
, the tumor capsule rupture is often present.
However
, this is not the negative predictive factor in itself. Please add ref. by Dr Raul
Ribeiro group.

13.
Could the Authors comment on two ref., regarding adrenal incidentalomas and
p
heochromocytoma: J. Clin. Med. 2021, 10, 5509; Endocrine 2021, 74, 76684.

Author Response

We thank the Reviewer for their valuable feedback, which has helped us improve the quality of the manuscript. Please find the point-by-point response below and the corrected copy of our manuscript, where changes have been highlighted in red link throughout the manuscript.

1. Please use “pheochromocytoma” instead of “pheocromocytoma”.

Reply: “pheocromocytoma has been changed to “pheochromocytoma” throughout the text.

2. Line 47: “lack of enhancement (change of <10 HU 47 between pre-and postcontrast imaging).”- it should be stated that this relates to CT imaging.

Reply: we have now added “CT” on line 48.

3. Lines 52-53: “By contrast, surgery may be considered in case an adrenal mass measures > 4 cm [4], and it is recommended in case an adrenal mass without diagnostic benign imaging features measures > 4 cm [2,4].- shouldn’t it be also <4 cm? If the adrenal mass does not present the benign features or even the radiological features of malignancy are present the indications for surgery include also tumours <4 cm. Maybe I do not understand correctly the context, please clarify.

Reply: This sentence refers to the size as a criterion for surgery and to the fact that surgery may be considered for masses > 4 cm, and it is indicated when masses > 4 cm do not have benign imaging features. We have rephrased the paragraph (lines 50-55) as follows: “masses without benign imaging features are candidates to further imaging and/or surgery. However, also size and growth are important variables in predicting malignancy, and they are generally used to make decisions regarding surgery or further imaging [2]. For instance, surgery may be considered in case an adrenal mass with benign imaging features measures > 4 cm [4], and it is recommended in case an adrenal mass without benign imaging features measures > 4 cm [2,4].

4. Lines 65-66: “including hypertension (persistent, paroxysmal or induced by medication), hypokalemia, diabetes mellitus, asymptomatic vertebral fractures, as well as hirsutism, virilization or gynecomastia”- lack of other common clinical characteristics of hypercortisolaemia.

Reply: Taking into account the Reviewer comment we have now added that “In addition, it is important to pay attention to other signs such as easy bruising, proximal myopathy, facial plethora, striae, which are the features that best discriminate Cushing’s syndrome [8,9].”, on page 2, lines 67-69.

5. Line 234: LMWH- please describe the abb.

Reply: we have clarified that LMWH refers to low molecular weight heparin (248-249)

6. Patient 2 and 4- what was the detailed histopathology of the tumours, according to Weiss/Weiss revisited criteria or other adequate system?

Reply: Taking into account the Reviewer comment, we have included the details of the pathology exam of case 2 “Morphological analysis (Weiss system [24]) showed small cells with round nuclei and no nucleoli, almost no mitoses, >25% clear cells, no venous, sinusoidal invasion or capsular infiltration. Ki67 staining showed a proliferation index <5%” (lines 243-246) and the details of the pathology exam of case 4 “final pathology showed an adrenal mass measuring 3.5x7.5x9.5 cm with a necrotic core of 6 cm containing hemorragic areas and no viable cells, surrounded by a peripheral rim of zona fasciculata. The complete absence of viable cells did not allow any morphological analysis nor any immunohistochemical staining. The exam was consistent with either an adrenal hematoma, or a nodule with a necrotic-hemorragic content.” (lines 404-409).

7. In my opinion the information in the paragraph: lines 272-276 is not very relevant. I understand that it was confirmed in the ref. However, this is the clear data from understanding the adrenal tumours biology. If this paragraph is to be left, I would recommend adding a comment that the cited ref. just confirm the highly suspected clinical scenario.

Reply: Taking into account the Reviewer comments, this part has been shortened.

8. Lines 303 and 381-382: the reference range to exclude autonomous cortisol secretion was presented before, there is no need to repeat the information.

Reply: The reference range has been deleted

9. Line 234: maybe cocaine, as its use mainly refers to drug abuse, may be presented at the end of the list of medications.

Reply: “cocaine” has been now moved at the end of the sentence

10. Lines 351-352: “In our patient, due to the size of the adrenal mass, the exam showed uptake of the tracer on the left adrenal gland only.”- unclear.

Reply: We apologize for the lack of clarity. What we meant (and failed to convey) was that we prescribed MIBG scintigraphy due to the size of the adrenal mass. The exam showed only adrenal MIBG uptake. We have removed the sentence, and rephrased the sentence on lines 327-329

11. Line 410: “In these cases, surgery should be performed by experienced surgeons (>15 adrenalectomies per year).”- please add ref.

Reply: The reference has now been added

12. Line 411: “Given that the only chance for cure in case of an adrenocortical carcinoma is complete primary tumor resection, with no tumor capsule rupture”- it is not always true. In paediatric adrenal tumors, due to fragile capsule, the tumor capsule rupture is often present. However, this is not the negative predictive factor in itself. Please add ref. by Dr Raul Ribeiro group.

Reply: We apologize with the Reviewer, but we have not been able to found the reference indicating that tumor capsule rupture is not a negative predictive factor.

13. Could the Authors comment on two ref., regarding adrenal incidentalomas and pheochromocytoma: J. Clin. Med. 2021, 10, 5509; Endocrine 2021, 74, 76–684.

Reply: We thank the Reviewer for this suggestion. The results of both studies have now been included in the manuscript on lines 261-265, and 379-383.

Reviewer 3 Report

In this review, Bernardi and colleagues are giving an overview on the work-up of patients with adrenal incidentalomas and illustrate their approach by discussing 4 exemplary patients.

Overall, the review is well written. I have the following comments:

1. On page 4 (lines 108-110) the authors write “In the case of our patient, aldosterone was 26.4 ng/dL (reference ranges 1.5-15 ng/dL) and renin was suppressed, being 0.5 μU/mL (reference ranges 2.8-40 μU/mL), with an aldosterone-to-renin-ratio (ARR) of 559”. With the units used, I find it difficult to follow the calculation. Wouldn’t it be better to use the unit ng/mL/hour for renin?

2. On page 6 (lines 205-206) the authors write “Potassium should be measured after surgery”. I agree. However, the authors do not provide the postoperative potassium level of the patient presented as case 1.

3. On page 8 (lines 313-315) the authors write “The patient underwent laparoscopic left adrenalectomy,…” and then “The patient underwent genetic testing,…”. Shouldn’t it rather be the other way around? How can cortical-sparing (cortex-sparing, partial or subtotal) adrenalectomy be performed (line 344) if the genetic information is not available preoperatively? If this is the rationale why the authors recommend to exclude MEN 2 (lines 331-332, Table 1 (please also see my comments below))? Why not excluded VHL and TMEM127 as well (see comment #Y)? In this regard, it seems incomplete to describe only MEN 2 under 4.2.

4. With regard to cortical-sparing adrenalectomy, the recommendation to perform it in patients with both MEN 2 and VHL should be widened by including TMEM127. A recent publication (Armaiz-Pena et al., JCEM 2021) showed that patients having a germline mutation in TMEM127 also have a relatively high risk (25%) of developing bilateral pheochromocytoma but a low risk (<3%) of malignancy. This is very similar to VHL.

5. Table 1: With regard to pheochromocytoma, I can’t follow the rational to exclude MEN 2 only. Genetic testing is recommended but why should only MEN 2 be excluded? The “chance” that a pheochromocytoma is diagnosed prior to medullary thyroid cancer in patients with MEN 2 is very low. And the pheochromocytomas would have to be operated first anyway. If the rational is cortical-sparing adrenalectomy, please widen your recommendation (see also comment #3).

6. Figure 1: There is a discrepancy between the legend for Figure 1f “…without extra-adrenal F-FDG uptake on subsequent PET-CT” and the text (lines 389-390) “…which showed moderate tracer uptake in the right 389 adrenal mass only”. It looks like no uptake at all.

7. I found a few typos. Line 238: “…it did not completely normalized” should be ““…it did not completely normalize”. Line 351: “…with knwn metastatic pheocromocytomas” should be “with known metastatic pheocromocytomas”. Line 390: “The patient underwent left laparotomic adrenalectomy…” should be “The patient underwent right laparotomic adrenalectomy…”.

Author Response

We thank the Reviewer for their valuable feedback, which has helped us improve the quality of the manuscript. The changes have been highlighted in red link.

1. On page 4 (lines 108-110) the authors write “In the case of our patient, aldosterone was 26.4 ng/dL (reference ranges 1.5-15 ng/dL) and renin was suppressed, being 0.5 μU/mL (reference ranges 2.8-40 μU/mL), with an aldosterone-to-renin-ratio (ARR) of 559”. With the units used, I find it difficult to follow the calculation. Wouldn’t it be better to use the unit ng/mL/hour for renin?

Reply: In order to use the unit ng/mL/h of renin we should have measured plasma renin activity rather than direct renin concentration, which are both accepted ways to evaluate any primary aldosteronism. The value of 559 depends on a calculation provided by the ARR-app, which takes into account aldosterone, either direct renin concentration or plasma renin activity, as well as potassium. This has been detailed on page 4, lines 114-118.

2. On page 6 (lines 205-206) the authors write “Potassium should be measured after surgery”. I agree. However, the authors do not provide the postoperative potassium level of the patient presented as case 1.

Reply: potassium has now been added on the case vignette #1, page 5, line 145.

3. On page 8 (lines 313-315) the authors write “The patient underwent laparoscopic left adrenalectomy,…” and then “The patient underwent genetic testing,…”. Shouldn’t it rather be the other way around? How can cortical-sparing (cortex-sparing, partial or subtotal) adrenalectomy be performed (line 344) if the genetic information is not available preoperatively?

Reply: We agree with the Reviewer that genetics should help decide if performing cortical-sparing adrenalectomy. Truth is, that in our case, the patient underwent the genetic testing before surgery, but then he was operated on before receiving the results due to a delay in the response and his willingness. Anyway, the test excluded genetic mutations and he underwent laparoscopic left adrenalectomy. In order to improve the linearity of the case vignette, the results of the genetic testing have been moved to lines 325-327.

4. If this is the rationale why the authors recommend to exclude MEN 2 (lines 331-332, Table 1 (please also see my comments below))? Why not excluded VHL and TMEM127 as well (see comment #Y)? In this regard, it seems incomplete to describe only MEN 2 under 4.2.

Reply: We thank the Reviewer for pointing out this aspect. We have now clarified that genetic disease includes MEN2A, MEN2B, von-Hippel Lindau (VHL), neurofibroma-tosis type 1 (NF1), as well as mutations in the genes of myc-associated factor-X ()MAX, trans-membrane protein 127 (TMEM127), and the succinate dehydrogenase complex (SDHx). In Table 1 we have removed MEN2 and left “genetic testing”.

5. With regard to cortical-sparing adrenalectomy, the recommendation to perform it in patients with both MEN 2 and VHL should be widened by including TMEM127. A recent publication (Armaiz-Pena et al., JCEM 2021) showed that patients having a germline mutation in TMEM127 also have a relatively high risk (25%) of developing bilateral pheochromocytoma but a low risk (<3%) of malignancy. This is very similar to VHL.

Reply: Taking into account the Reviewer comment, we have included TMEM127 mutations, the reference to the work by Armaiz-Pena et al, as well as the concept that TMEM127 mutations and VHL disease infrequently cause metastases, lines 349-358.

6. Table 1: With regard to pheochromocytoma, I can’t follow the rational to exclude MEN 2 only. Genetic testing is recommended but why should only MEN 2 be excluded? The “chance” that a pheochromocytoma is diagnosed prior to medullary thyroid cancer in patients with MEN 2 is very low. And the pheochromocytomas would have to be operated first anyway. If the rational is cortical-sparing adrenalectomy, please widen your recommendation (see also comment #3).

Reply: The recommendation has been widened and we have now specified that “Genetic disease includes MEN2A, MEN2B, von-Hippel Lindau (VHL), neurofibromatosis type 1 (NF1), as well as mutations in the genes of myc-associated factor-X (MAX), trans-membrane protein 127 (TMEM127)[36], and the succinate dehydrogenase complex (SDHx)”.

7. Figure 1: There is a discrepancy between the legend for Figure 1f “…without extra-adrenal F-FDG uptake on subsequent PET-CT” and the text (lines 389-390) “…which showed moderate tracer uptake in the right 389 adrenal mass only”. It looks like no uptake at all.

Reply: We apologize for the mistake and to amend this discrepancy, we have now written “no extra-adrenal 18F-FDG uptake”, like in the Figure legend.

8. I found a few typos. Line 238: “…it did not completely normalized” should be ““…it did not completely normalize”. Line 351: “…with knwn metastatic pheocromocytomas” should be “with known metastatic pheocromocytomas”. Line 390: “The patient underwent left laparotomic adrenalectomy…” should be “The patient underwent right laparotomic adrenalectomy…”.

Reply: We thank the Reviewer for pointing these typos out.